# Clinical Implications of the *FLT3*-ITD Allelic Ratio in Acute Myeloid Leukemia in the Context of an Allogeneic Stem Cell Transplantation

**DOI:** 10.3390/cancers15041312

**Published:** 2023-02-18

**Authors:** Madlen Jentzsch, Lara Bischof, Dominic Brauer, Donata Backhaus, Jule Ussmann, Georg-Nikolaus Franke, Vladan Vucinic, Uwe Platzbecker, Sebastian Schwind

**Affiliations:** Medical Clinic and Policlinic 1, Hematology, Cellular Therapy and Hemostaseology, University of Leipzig Medical Center, 04103 Leipzig, Germany

**Keywords:** acute myeloid leukemia, *FLT3*, allelic ratio, hematopoietic stem cell transplantation, prognosis

## Abstract

**Simple Summary:**

The presence of *FLT3*-ITD is among the most common molecular aberrations in acute myeloid leukemia (AML). Although patients harboring *FLT3*-ITD are often consolidated by allogeneic hematopoietic stem cell transplantation (HSCT), little is known about how the *FLT3*-ITD allelic ratio impacts patient outcomes after HSCT. Here, we analyzed the biological and clinical features of these patients in the context of other risk factors, including the ELN2017 risk and the measurable residual disease status (MRD) at HSCT. Our data shows no survival differences between patients with a high or low *FLT3*-ITD allelic ratio in the context of an allogeneic HSCT, but highlights the importance of pre-HSCT MRD as a prognostic factor.

**Abstract:**

Although the presence of *FLT3*-ITD, as well as levels of the *FLT3*-ITD allelic ratio, have been described as prognostic factors in acute myeloid leukemia (AML), little is known about how the *FLT3*-ITD allelic ratio impacts patients’ outcomes when receiving an allogeneic hematopoietic stem cell transplantation (HSCT). We analyzed 118 patients (median age at diagnosis 58.3, range 14.3–82.3 years) harboring *FLT3*-ITD, of whom 94 patients were consolidated with an allogeneic HSCT and included in outcome analyses. A high *FLT3*-ITD allelic ratio was associated with a higher white blood cell count, higher blood and bone marrow blasts, and worse ELN2017 risk at diagnosis. Patients with a high *FLT3*-ITD allelic ratio more often had *NPM1* mutations, while patients with a low allelic ratio more often had *FLT3*-TKD mutations. Patients with a high *FLT3*-ITD allelic ratio were less likely to achieve a measurable residual disease (MRD)-negative remission prior to allogeneic HSCT and had a trend for a shorter time to relapse. However, there was no distinct cumulative incidence of relapse, non-relapse mortality, or overall survival according to the *FLT3*-ITD allelic ratio in transplanted patients. While co-mutated *FLT3*-TKD was associated with better outcomes, the MRD status at HSCT was the most significant factor for outcomes. While our data indicates that an allogeneic HSCT may mitigate the adverse effect of a high *FLT3*-ITD allelic ratio, comparative studies are needed to evaluate which *FLT3*-ITD mutated patients benefit from which consolidation strategy.

## 1. Introduction

In patients with acute myeloid leukemia (AML), the presence of internal tandem duplication (ITD) in the Fms-like tyrosine kinase (*FLT3*) gene has been associated with similar induction success as wild-type *FLT3*, but much shorter remission duration, higher likelihood of early relapse and shorter overall survival (OS) [1,2,3]. Additionally, the *FLT3*-ITD mutational burden, usually measured as an allelic ratio, impacts outcome [3,4,5], which seems to be further altered by the mutational status of the *FLT3* tyrosine kinase domain (*FLT3*-TKD) with inconsistent data [6,7] and favorably by the *NPM1* gene co-mutational status [8,9,10,11]. Subsequently, the *FLT3*-ITD allelic ratio (with a cut at 0.5), together with the *NPM1* mutation status, were incorporated in the European LeukemiaNet (ELN) 2017 risk stratification [12].

To engage the oncogenic signaling arising from mutated *FLT3*, a number of tyrosine kinase inhibitors (TKI) were developed. In newly diagnosed *FLT3*-ITD or *FLT3*-TKD mutated AML, the RATIFY trial demonstrated that the addition of midostaurin to standard induction therapy prolonged OS [13], also independently from the *NPM1* mutation/*FLT3*-ITD allelic ratio combinations [8], resulting in midostaurin as a new standard of care. Overall, this survival benefit was particularly seen in patients receiving allogeneic stem cell transplantation (HSCT) in first complete remission (CR) [13]. Unfortunately, we still lack prospective randomized trials evaluating the optimal post-remission therapeutic strategy in *FLT3*-mutated AML, also taking into consideration the *FLT3*-ITD allelic ratio, co-mutation combinations, and—more recently—the use of FLT3 inhibitors during treatment. Additional to the genetic risk at diagnosis, the measurable residual disease (MRD) status during chemotherapy allows further risk stratification and impacts treatment decisions on consolidation therapy. While a variety of mutations may function as MRD markers in AML [14,15,16,17], *NPM1* mutation-based MRD remains the most accepted MRD method and has been included in the algorithms for consolidating *FLT3*-ITD mutated patients [18,19,20]. As a consequence of the conflicting results of gathered data, the optimal consolidation in *FLT3*-ITD mutated AML remains a matter of debate, especially in patients with a low allelic ratio. One German analysis indicated no benefit of an allogeneic HSCT in first CR for patients with a low allelic ratio without considering *NPM1* mutation status [21], while two other studies found OS and event-free survival (EFS) only improved by allogeneic HSCT in low allelic ratio patients without a concomitant *NPM1* mutation [22,23]. In contrast, several other investigations found outcome improvements with HSCT for *FLT3*-ITD regardless of the allelic ratio, or the *NPM1* mutation status [24,25,26,27]. Subsequently, the indication for HSCT in the first CR in patients with an *FLT3*-ITD allelic ratio < 0.5 and a concomitant *NPM1* mutation who achieve MRD-negativity remains controversial between the National Comprehensive Cancer Network (NCCN) guidelines advocating HSCT in CR [28] and ELN recommending a delay until first relapse [12]. For all other patients with *FLT3*-ITD mutated AML, an allogeneic HSCT in CR1 should be strongly considered [29,30]. Even though HSCT has a significant role in consolidating *FLT3*-ITD positive AML patients [12,30], no study analyzed the role of the allelic ratio in this context, leaving substantial knowledge gaps concerning factors associated with outcomes following HSCT in these patients [30]. Therefore, in this study, we analyzed *FLT3*-ITD positive AML patients concerning the allelic ratio, co-mutational status, MRD burden, and outcomes after consolidating allogeneic HSCT.

## 2. Subjects and Methods

### 2.1. Patients and Treatment

A total of 118 patients diagnosed with an *FLT3*-ITD positive AML between 2001 and 2018 at a median age of 58.3 (range 14.3–82.3 years) were included for association analyses. Of those, 80% (*n* = 94) received an allogeneic HSCT and were included in outcome analyses. Further clinical and biological characteristics are given in Table 1 and Appendix A. Conditioning regimens for the patients included in the outcome analyses were either myeloablative (MAC, *n* = 32, 34%), of reduced intensity (RIC, *n* = 15, 16%) or non-myeloablative (NMA, *n* = 47, *n* = 50%). Reasons for NMA-HSCT or RIC-HSCT as opposed to MAC-HSCT were age over 50 years if receiving unrelated HSCT and over 55 years if receiving related HSCT, prior autologous HSCT (*n* = 3) or active infections (*n* = 2). All patients received granulocyte-colony stimulating factor-mobilized peripheral blood stem cells as a graft source. Further HSCT-related characteristics and details on the applied chemotherapy protocols prior to allogeneic HSCT are given in Appendix A, respectively. The median follow-up after HSCT was 2.7 years for living patients. Written informed consent was obtained from all patients in accordance with the Declaration of Helsinki.

### 2.2. Use of FLT3 Inhibitors Prior to and after Allogeneic HSCT

The use of FLT3 inhibitors was dependent on the standard of care at the time of diagnosis. Most patients were diagnosed prior to the approval of midostaurin in Germany, leading to the restricted use of the substance in this patient cohort. Five patients were treated with an FLT3 inhibitor (four patients received induction with 7 + 3 and midostaurin and one patient received gilteritinib after non-response to standard induction chemotherapy within the ADMIRAL trial) [13,31]. Seven additional patients were included in a randomized double-blinded trial that evaluated the efficacy of Quizartinib (ClinicalTrials.gov Identifier: NCT02668653), resulting in a 50% chance of receiving the substance. Three of those seven patients received maintenance therapy after HSCT within the study.

### 2.3. Analyses of Cytogenetics, Molecular Data, and Immunophenotype

Pretreatment bone marrow cytogenetic analyses were performed centrally using standard techniques of banding and in situ hybridization. Genomic DNA of pretreatment samples were screened for the presence of the *CEBPA*, *FLT3*-ITD, *FLT3*-TKD, and *NPM1* mutations, as well as of genes included in the Myeloid Panel by Illumina (San Diego, CA, USA), as previously described [32,33,34]. The *FLT3*-ITD allelic ratio was determined by dividing the area under the curve of the mutant peak by that of the wild-type *FLT3*-ITD peak. In cases with >1 detectable *FLT3*-ITD, all values were added up to one value. For all analyses, patients were grouped according to the allelic ratio with a cut at 0.5 as suggested by the ELN2017 risk stratification [12]. Patients were grouped according to the ELN2017 genetic risk classification [12]. Determination of flow cytometry results and the leukemic stem cell population at diagnosis was performed as previously described [33].

### 2.4. MRD Analysis

For patients transplanted in CR or CR with incomplete peripheral count recovery (CRi) with material available, the pre-HSCT MRD status was analyzed by custom-made digital droplet PCR primer/probe assays for the detection of *NPM1* as previously described [19] and *FLT3*-TKD as shown in the Appendix A. Of the 35 patients analyzed, MRD relied on *NPM1* alone in 29 patients, on *FLT3*-TKD alone in three patients, and on both *NPM1* and *FLT3*-TKD in three patients. Of the three patients with MRD results for both genes, all were congruently MRD negative.

### 2.5. Definition of Clinical Endpoints and Statistical Analyses

Statistical analyses were performed using version 3.4.3 of the R statistical software platform [35]. OS and EFS were calculated from HSCT until AML-relapse or death from any cause. Survival estimates were calculated using the Kaplan-Meier method and groups were compared using the log-rank test. The competing risks of cumulative incidence of relapse/progression (CIR) and non-relapse mortality (NRM) were calculated from HSCT to relapse or death, respectively, using the Fine and Gray method [36]. Associations with baseline clinical, demographic, and genetic features were compared using the Kruskal-Wallis-Test and Fisher’s exact tests for continuous and categorical variables, respectively. Multivariate analysis is described in the Appendix A.

## 3. Results

### 3.1. Association of the FLT3-ITD Allelic Ratio at Diagnosis

Compared to patients with a low *FLT3*-ITD allelic ratio, patients with a high *FLT3*-ITD allelic ratio had a lower platelet count (*p* = 0.03, Table 1) as well as a higher white blood cell count (*p* < 0.001) and higher blood (*p* < 0.001) and marrow (*p* < 0.001) blast percentages at diagnosis. They also showed a higher incidence of worse and a lower incidence of favorable risk according to ELN2017 (*p* = 0.02) and presented with a higher CD34+/CD38− cell burden (*p* < 0.001), a higher CD33 expression (*p* < 0.001, Figure 1) and a distinct immunophenotype at diagnosis, which is described in the Appendix A). Regarding the co-mutational status, we observed a higher incidence of *NPM1* mutations (*p* = 0.005) in patients with a high *FLT3*-ITD allelic ratio as well as a higher incidence of *FLT3*-TKD mutations in patients with a low *FLT3*-ITD allelic ratio (*p* = 0.007, Figure 2A).

### 3.2. Outcome According to the FLT3-ITD Allelic Ratio at Diagnosis

Patients with a high *FLT3*-ITD allelic ratio were less likely to achieve MRD-negative remission prior to allogeneic HSCT (*p* = 0.02, Appendix A), and—in patients suffering relapse after HSCT—there was also a trend for shorter time to relapse for high *FLT3*-ITD allelic ratio patients (*p* = 0.06, Figure 2B).

However, patients with a high or low *FLT3*-ITD allelic ratio at diagnosis did not differ regarding their CIR (*p* = 0.57, Figure 2C), NRM (*p* = 0.82, Appendix A), or OS (*p* = 0.50, Figure 2D) when receiving an allogeneic HSCT. Similar results were obtained when we analyzed the time from diagnosis to relapse or death (Appendix A). We also did not observe a prognostic impact of the *FLT3*-ITD allelic ratio when additional cut-offs at 0.25 and 0.75 were introduced (Appendix A) or when we excluded all patients (potentially) receiving an FLT3 inhibitor prior to allogeneic HSCT (Appendix A).

### 3.3. Outcomes in the Context of NPM1 or FLT3-TKD Mutation Status

As we observed a significant distinct distribution of the *NPM1* and *FLT3*-TKD mutation status according to the *FLT3*-ITD allelic ratio at diagnosis, outcomes were analyzed in the context of these markers. In the whole *FLT3*-ITD mutated patient cohort, the *NPM1* co-mutational status did not significantly impact any analyzed endpoint (EFS *p* = 0.30, Figure 3A; CIR *p* = 0.19 and OS *p* = 0.40, Appendix A). Although *NPM1* mutated AML patients with a low *FLT3*-ITD allelic ratio tended to have a lower CIR compared to all other patients (*p* = 0.09), there were no significant outcome differences according to the *NPM1* mutation and *FLT3*-ITD allelic ratio combinations (EFS *p* = 0.60, Figure 3B; CIR *p* = 0.36 and OS *p* = 0.60, Appendix A).

With regard to the *FLT3*-TKD mutation status in all transplanted *FLT3*-ITD mutated patients, we observed a significantly longer EFS (*p* = 0.02, Figure 3C) as well as a trend for a lower CIR and longer OS (*p* = 0.12 and *p* = 0.06, respectively, Appendix A). A separate analysis according to the *FLT3*-ITD allelic ratio and the *FLT3*-TKD mutation status showed this to be especially due to the favorable outcomes of patients with a low *FLT3*-ITD allelic ratio and mutated *FLT3*-TKD (EFS *p* = 0.03, Figure 3D; CIR *p* = 0.19 and OS *p* = 0.30, Appendix A).

### 3.4. Outcomes According to NPM1 and FLT3-TKD Based MRD Analysis

The detection of persistent *NPM1* or *FLT3*-TKD mutation was a strong predictor for higher CIR (*p* < 0.001, Figure 4A), as well as significantly shorter OS (*p* = 0.03, Figure 4B). In addition to the established MRD marker *NPM1* [19,37], persistent *FLT3*-TKD also provided important prognostic information in remission: all six patients for which *FLT3*-TKD MRD analysis was performed tested MRD negative and were alive and in remission during the last follow up (median 2.5 years after HSCT).

## 4. Discussion

Due to the lack of randomized adequately powered trials and conflicting results of outcome data, there is still no final consensus regarding the optimal consolidation treatment for each patient harboring *FLT3*-ITD. Moreover, the *FLT3*-ITD allelic ratio threshold remains a matter of debate, as the *FLT3*-ITD allelic ratio shows a continuous impact on disease aggressiveness [38]. The introduced thresholds differed between published studies (ranging from 0.25 [5] to 0.80 [22]) and usually relied on the median value with the now broadly used 0.5 cuts being a result of the maximum of prognostic data. While patients with a high *FLT3*-ITD allelic ratio with or without mutated *NPM1* (i.e., intermediate or adverse risk patients according to ELN2017) are usually subjected to allogeneic HSCT in first CR, clinical debate remains on the optimal consolidation for low *FLT3*-ITD allelic ratio patients, especially with concurrent *NPM1* mutations [20]. In their recently published position statement paper, the acute leukemia working party of the EBMT recommended allogeneic HSCT in first CR for all *FLT3*-ITD mutated AML patients with the exception of *NPM1* mutated/low *FLT3*-ITD allelic ratio individuals achieving an MRD-negative CR [30]. This mostly relies on retrospective analyses indicating dismal outcomes of *FLT3*-ITD-positive patients not undergoing allogeneic HSCT, irrespective of the observed allelic ratio [5,24,26,39]. However, data analyzing the outcomes of high vs. low *FLT3*-ITD allelic ratio in the context of an allogeneic HSCT have been lacking [30].

To our best knowledge, we are the first to analyze *FLT3*-ITD high vs. low allelic ratio mutated patients regarding their co-mutations, as well as outcomes after allogeneic HSCT. First, we noted a strong correlation of a higher *FLT3*-ITD allelic ratio (as a continuous variable as well as after introducing a 0.5 allelic ratio cuts) with mutated *NPM1* and of lower *FLT3*-ITD allelic ratio with mutated *FLT3*-TKD (Figure 2A). Similar to the known associations of *FLT3*-ITD with higher white blood count and higher bone marrow and blood blast percentages [1,3] we observed these associations for patients with a high vs. a low *FLT3*-ITD allelic ratio. There also was a distinct immunophenotype according to the *FLT3*-ITD allelic ratio with a higher expression of myeloid differentiation antigens and lower expression of T-cell, erythroid, and platelet differentiation antigens in patients with a high *FLT3*-ITD allelic ratio (Appendix A). Particularly, patients with a high *FLT3*-ITD allelic ratio had a higher CD33 expression, which was also dependent on the *NPM1* mutation status, resulting in the highest CD33 expressions in high *FLT3*-ITD allelic ratio/mutated *NPM1* and lowest in *FLT3*-ITD allelic ratio/wild-type *NPM1* (*p* < 0.001, Figure 1). Subsequently, there may be a potential clinical benefit in adding the CD33 inhibitor Gemtuzumab ozogamizin also in patients with a high *FLT3*-ITD allelic ratio, especially when they harbor an additional *NPM1* mutation, a strategy that is currently evaluated in clinical studies (ClinicalTrials.gov Identifier: NCT04385290, NCT03900949).

Despite a higher portion of patients with a high *FLT3*-ITD allelic ratio remaining MRD-positive prior to HSCT, a high *FLT3*-ITD allelic ratio only associated with a trend for a shorter time to relapse in relapsing patients, while CIR and OS did not differ from patients with a low *FLT3*-ITD allelic ratio. Importantly, OS rates five years after allogeneic HSCT were similarly high with 55% and 58% in patients with a high or a low *FLT3*-ITD allelic ratio, respectively. Although the comparison to previous studies has to be interpreted with caution, these results stand in line with outcomes of transplanted patients in the study of Ho et al. [22] who achieved a five year OS of approximately 50% and 55% in high and low allelic ratio patients, respectively. Despite a significantly older age in our cohort (58 vs. 48 years), patients’ outcomes in our study compared favorably to the Simon-Makuch analysis of Schlenk et al., which stated no benefit from allogeneic HSCT in low *FLT3*-ITD allelic ratio patients [21]. Here, patients with a low *FLT3*-ITD allelic ratio only achieved five year OS rates of approximately 40%, irrespective of consolidating chemotherapy or allogeneic HSCT. In our study, five year OS was longer in both, patients with a high or a low *FLT3*-ITD allelic ratio, probably as NRM rates after HSCT in our analysis were as low as 13% at five years (Appendix A).

As the conditioning intensity was heterogeneous in our study, we performed separate analyses of patients receiving NMA/RIC or MAC conditioning (Appendix A). In patients after NMA/RIC, but not after MAC conditioning, there was a non-significant optical separation of the outcome curves, which may indicate a potential benefit for patients with a high *FLT3*-ITD from more intensive conditioning regimens. However, patient numbers were small and additional studies are needed to further evaluate this point.

In addition to the *FLT3*-ITD allelic ratio, the co-mutational status of other AML-associated genes as well as the MRD status prior to allogeneic HSCT, may impact outcomes. Despite relatively low relapse risk following HSCT in the ELN2017 favorable risk combination mutated *NPM1* and low *FLT3*-ITD AR (against all others, *p* = 0.09), neither the *NPM1* mutation status alone nor the combination with the *FLT3*-ITD allelic ratio significantly impacted outcomes after allogeneic HSCT (Figure 3A,B and Appendix A). In contrast, the presence of mutated *FLT3*-TKD in addition to *FLT3*-ITD showed very favorable outcomes after allogeneic HSCT, especially in patients with a low *FLT3*-ITD allelic ratio (Figure 3C,D and Appendix A). While our study is the first to exclusively analyze *FLT3*-ITD mutated patients, others already suggested improved outcomes in *FLT3*-TKD mutated AML patients, either alone or when co-mutated *NPM1* was present [6,40], but this was not consistent across all reports [7].

As expected, there was a strong prognostic impact of the pre-transplant MRD status on all analyzed endpoints (Figure 4), which was also confirmed in a multivariate analysis (Appendix A). Although patient numbers were low, our data indicated that the prognostic relevance of the MRD status was independent of the *FLT3*-ITD allelic ratio. Of the two MRD-positive patients with a low *FLT3*-ITD allelic ratio, both died within a year after HSCT, one after early relapse and one from treatment-related complications. Of the nine patients with high *FLT3*-ITD allelic ratio transplanted in MRD-negative remission, only one relapsed after HSCT. Subsequently, the MRD status at HSCT, rather than the allelic ratio or co-mutational status at diagnosis—remained the most relevant prognostic factor in *FLT3*-ITD mutated AML patients undergoing allogeneic HSCT. This is in line with the suggestions of the new ELN risk classification of 2022, in which *FLT3*-ITD defines intermediate risk, irrespective of the *NPM1* co-mutation status, and MRD analyses at informed time points during disease course are recommended to dynamically adjust disease risk of favorable and intermediate risk patients [41].

The introduction of a variety of new substances is already further adjusting the care of *FLT3*-ITD mutated patients. The approval of midostaurin as a combination partner to standard 7 + 3 including a maintenance phase after chemotherapy provides an alternative to consolidating allogeneic HSCT in selected patients and can be guided by serial MRD monitoring [13,42]. For another FLT3 inhibitor, sorafenib, convincing randomized study results were published in the context of maintenance therapy after allogeneic HSCT [43,44]. Two independent studies showed a reduced relapse rate and prolonged OS for patients receiving the substance, which can further improve outcomes of this high-risk patient population.

Despite the increasingly promising outcomes of *FLT3*-ITD mutated patients after allogeneic HSCT, relapse remains a major clinical problem, which also occurred in 48% of patients at five years in our analysis, irrespective of the *FLT3*-ITD allelic ratio. Recently, monotherapy with the second-generation FLT3 inhibitor gilteritinib was approved for the treatment of relapsed/refractory *FLT3* mutated AML in the US and Europe [31]. Practice-changing importance of FLT3 inhibitors in relapsing *FLT3*-positive patients may lie in a realistic chance to achieve a second CR which—as a bridge to transplant—can still result in long-term outcomes. Finally, the clinical benefit of FLT3 inhibitors may also be more pronounced in patients with a high *FLT3* allelic ratio, higher therapy intensity, and *NPM1* co-mutation [45].

Our study has some limitations, including limited patient numbers in subgroup analyses and infrequent and heterogeneous use of FLT3 inhibitors, preventing conclusions in the context of these new substances which most likely will continue to further improve outcomes. However, results remained consistent when we excluded patients treated with FLT3 inhibitors prior to allogeneic HSCT (Appendix A). Additionally, we cannot comment on the outcomes of patients consolidated with chemotherapy (including FLT3-targeted maintenance), preventing final conclusions on the optimal consolidation treatment of patients harboring *FLT3*-ITD.

## 5. Conclusions

In conclusion, our study is the first to address a significantly different distribution of *FLT3*-TKD and *NPM1* mutations according to the *FLT3*-ITD allelic ratio as well as outcomes in the context of these mutations. Patients with a high or low *FLT3*-ITD allelic ratio had similar outcomes after allogeneic HSCT, indicating the potential of an allogeneic HSCT to mitigate the adverse prognostic impact of a high *FLT3*-ITD allelic ratio in AML. Finally, the pre-transplant MRD status remained the most important prognostic factor for outcomes of *FLT3*-ITD AML patients after allogeneic HSCT.

## Figures and Tables

**Figure 1 cancers-15-01312-f001:**
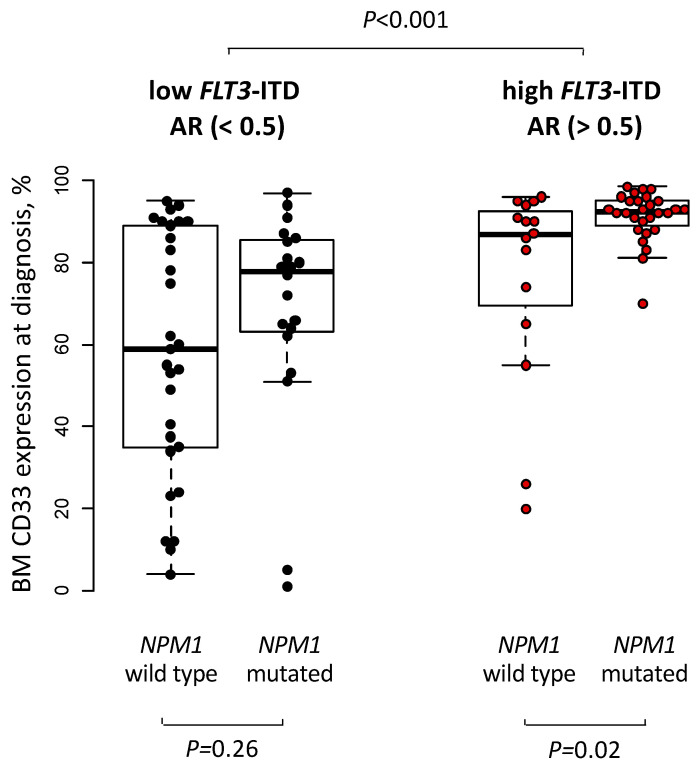
Bone Marrow CD33 expression levels in the context of *FLT3*-ITD allelic ratio (AR) as well as *NPM1* mutation status.

**Figure 2 cancers-15-01312-f002:**
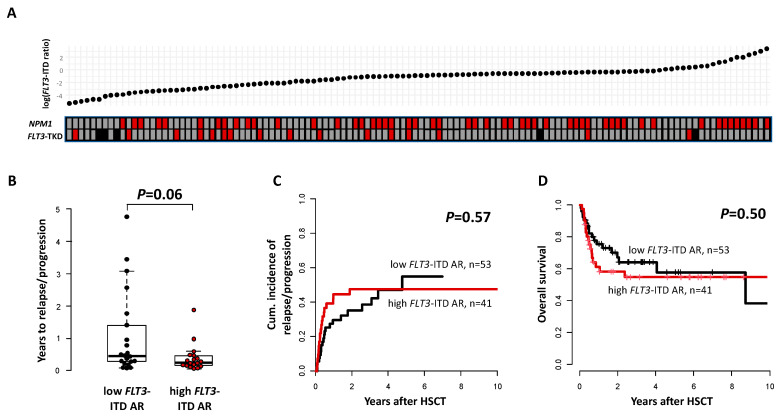
Associations and Outcome according to *FLT3*-ITD allelic ratio (AR, high vs. low, 0.5 cuts, *n* = 94). (**A**) Association between *FLT3*-ITD allelic ratio and *NPM1* as well as *FLT3*-TKD mutation status. Grey: wild-type, red: mutated, black: missing information. (**B**) Time to relapse in relapsing patients, (**C**) Cumulative incidence of relapse/progression, and (**D**) Overall survival.

**Figure 3 cancers-15-01312-f003:**
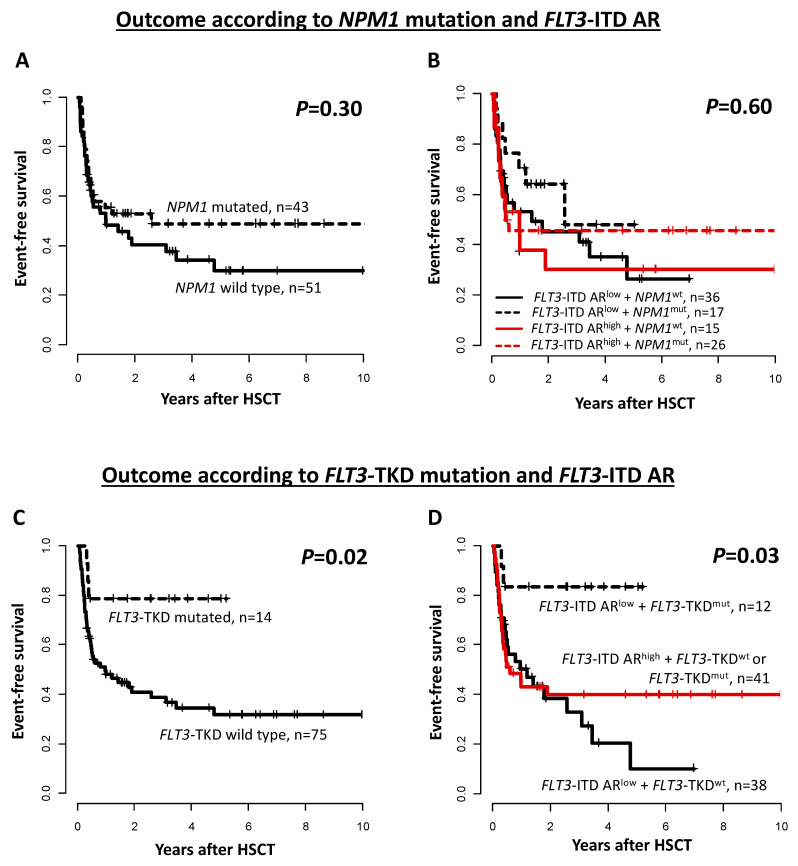
Event-free survival of patients harboring an *FLT3*-ITD undergoing allogeneic HSCT according to (**A**) the mutational status of *NPM1*, and (**B**) according to the mutational status of *NPM1* and the *FLT3*-ITD allelic ratio (AR, high vs. low, 0.5 cuts, *n* = 94), (**C**) the mutational status of *FLT3*-TKD, and (**D**) according to the mutational status of *FLT3*-TKD and the *FLT3*-ITD AR (high vs. low, 0.5 cuts, *n* = 89).

**Figure 4 cancers-15-01312-f004:**
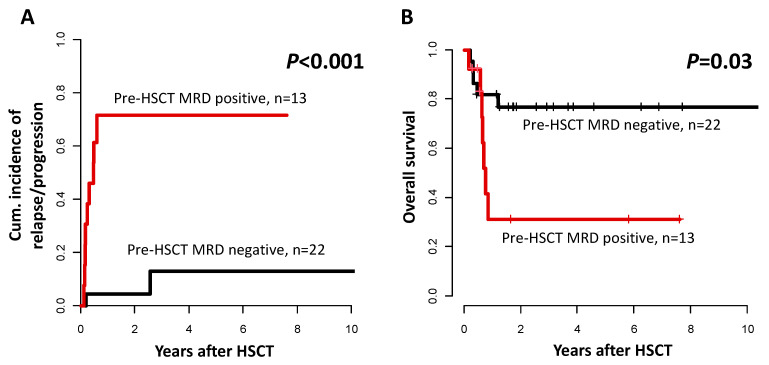
Outcomes according to *NPM1*, and *FLT3*-TKD MRD status prior to allogeneic HSCT in patients transplanted in remission (*n* = 35). (**A**) Cumulative incidence of relapse/progression and (**B**) Overall survival.

**Table 1 cancers-15-01312-t001:** Clinical and genetic characteristics for all patients according to *FLT3*-ITD allelic ratio at diagnosis (high vs. low, 0.5 cuts), *n* = 118.

	All Patients *n* = 118	Low *FLT3*-ITD AR *n* = 67	High *FLT3*-ITD AR *n* = 51	*p*
Age at diagnosis, years				0.41
median	58.3	57.5	60.5	
range	14.3–82.3	14.3–80.0	23.3–82.3	
Sex, *n* (%)				0.19
male	51 (43)	25 (37)	26 (51)	
female	67 (57)	42 (63)	25 (49)	
Disease origin, *n* (%)				0.62
secondary	19 (16)	12 (18)	7 (14)	
de novo	99 (84)	55 (82)	44 (86)	
Hemoglobin, g/dL				0.82
median	9.0	8.9	9	
range	5.3–13.5	5.6–13.5	5.3–13.2	
Platelet count, ×10^9^/L				0.03
median	63	71	54	
range	7–313	7–289	9–313	
WBC, ×10^9^/L				<0.001
median	22.7	9.3	70.7	
range	0.6–98	0.6–146	0.7–385	
Blood blasts, %				<0.001
median	50	27	76	
range	0–98	0–96	0–98	
BM blasts, %				<0.001
median	75	62	80	
range	4.6–95	4.6–90	25–95	
BM CD34+/CD38− burden, %				<0.001
median	1	0.3	2	
range	0–75	0–75	0–33	
BM CD33 expression, %				<0.001
median	86	66	92	
range	1–99	1–97	20–99	
Normal karyotype, *n* (%)				0.41
absent	36 (33)	23 (37)	13 (28)	
present	74 (67)	40 (63)	34 (72)	
ELN2017 genetic risk group, *n* (%)				<0.001
favorable	31 (28)	30 (50)	1 (2)	
intermediate	50 (46)	24 (38)	27 (55)	
adverse	28 (26)	7 (12)	21 (43)	
*NPM1*, *n* (%)				0.005
wild-type	60 (51)	42 (63)	18 (35)	
mutated	58 (49)	25 (37)	33 (65)	
*CEBPA*, *n* (%)				0.73
wild-type	89 (9)	51 (89)	38 (93)	
mutated	9 (91)	6 (11)	3 (7)	
*FLT3*-TKD, *n* (%)				0.007
wild-type	97 (86)	50 (78)	47 (96)	
mutated	16 (14)	14 (22)	2 (4)	

Abbreviations: AR, allelic ratio; BM, bone marrow; ELN, European LeukemiaNet, WBC, white blood cell count.

## Data Availability

Data presented in this study may be available upon request from the corresponding author.

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
