# Peer review of "Clinical Implications of the FLT3-ITD Allelic Ratio in Acute Myeloid Leukemia in the Context of an Allogeneic Stem Cell Transplantation"

_cancers, 2023, doi:10.3390/cancers15041312_

Round 1

Reviewer 1 Report

Jentzsch et al. present a retrospective study of FLT3-ITD mutated AML patients who received transplantation. The data presented here suggests that FLT3-ITD allelic ratio (high AR vs low AR) does not predict for survival in FLT3-ITD mutated AML patients after transplant. This finding is consistent with data from prior studies (eg. see Schlenk et al. 2014, Blood) which have shown a non-significant impact of FLT3-ITD allelic ratio in transplanted AML patients, thus this study has limited novelty.

Major comments:

1.    There are several limitations of this study. First, patient group numbers are relatively small in some subsets, so some analyses lack sufficient statistical power. Second, the retrospective nature of the study limits the significance of the survival analysis due to heterogeneous treatment. The authors should mention these limitations in the Discussion section.

2.      Patient numbers in the 2017 ELN risk groups should be carefully re-checked. Table 1 indicates that in the high FLT3-ITD AR group, one patient was listed as favorable risk. According to 2017 ELN risk classification, high FLT3-ITD AR would result in intermediate/high-risk categorization.  

3.      Table 1, BM blasts, %: the ranges are from 4.6 to 95%. For those BM blasts less than 20%, how the authors determine they are AMLs? Assume authors still follow the WHO 2017, and the blast percentage 20% cutoff only apply to recurrent genetic abnormalities t(15;17), t(8;21), inv(16), or t(16;16). (Hartmut Döhner et al; Diagnosis and management of AML in adults: 2017 ELN recommendations from an international expert panel. Blood 2017; 129 (4): 424–447. doi: https://doi.org/10.1182/blood-2016-08-733196)

4.      “Subsequently, with higher 5-year OS irrespective of the FLT3-ITD allelic ratio in our study, our data point to a potential usefulness of allogeneic HSCT also in patients with a low FLT3-ITD allelic ratio” and “Our data shows no survival differences between patients with a high or low FLT3-ITD allelic ratio in the context of an allogeneic HSCT, strengthening its use as consolidation option in these patients.” These conclusions regarding the usefulness of HSCT in patients with low FLT3-ITD AR are controversial and this study is not designed to evaluate whether HSCT is beneficial in patients with low FLT3-ITD AR. The appropriate control group to assess this conclusion is non-transplanted patients with low FLT3-ITD AR, and the relevant analysis is to compare outcomes of patients with low FLT3-ITD AR in terms of transplant status. Comparing their outcomes to patients with high FLT3-ITD AR does not address this issue.

5.      “Although patient numbers were low, our data indicated that the prognostic relevance of the MRD status was independent from the FLT3-ITD allelic ratio.” Multivariable analysis should be provided to determine whether a risk factor is prognostically independent of other risk factors. “independent” can be replaced with “irrespective” in this situation.

6.      The data presented here should be contextualized with regard to the 2022 ELN guidelines for AML. Importantly, FLT3-ITD AR is no longer considered for risk classification purposes. Furthermore, there is increasing support for the use of MRD status rather than FLT3-ITD AR to determine whether patients should go on to transplant. The 2022 ELN guidelines state: “allogeneic HCT is recommended for patients with adverse-risk AML and for the majority of those with intermediate-risk disease, although quite a few centers rely on the presence of MRD to guide their decision based on the predicted risk of relapse.” The authors should discuss these issues in relation to new ELN guidelines.

7.      For the outcome analysis, the authors seem to only calculate the time from transplant to the endpoint. The authors may indicate if there are differences from time of diagnosis to the endpoint.

8.      The authors may provide some explanation with regard to the fewer number of Midostaurin use in their cohort, especially there was none in the FLT3-ITD AR high group (5 were blinded).

Minor comments:

1.      Line 32, “A high FLT3-ITD allelic ratio associated with…” should be “A high FLT3-ITD allelic ratio is associated with…”

2.      Line 50, redundant word “also”: Additionally, also the FLT3-ITD mutational burden, usually measured as allelic ratio impacts outcome

3.      Line 72, grammatical error “german” should be changed to “German”

4.      Line 82, “Guidelines advocating HSCT in CR28 and ELN recommending a delay until first re- 82 lapse.” Incomplete sentence.

5.      Table 1, “Disease origin, n(%), secondary, de novo, all patients, 19, 99” should also include the percentages.

6.      Line 139, grammatical error “until or”: OS and EFS were calculated from HSCT until or AML-relapse or death from any cause.

7.      Line 149 to 158, in this paragraph, the words “allelic ratio” are Arial font and are different with the rest which are Times New Roman.

8.      Figure 1 B. It will be more consistent throughout the article if authors can set the “years to relapse/progression” as Y axis, and set the two groups as X axis.

9.      Line 190 (Figure legend), Letters A and B are repeated twice, order should follow A, B, C, D: Figure 3. Event-free survival of patients harboring a FLT3-ITD undergoing allogeneic HSCT ac-190 cording to (A) the mutational status of NPM1 and (B) according to the mutational status of NPM1 191 and the FLT3-ITD allelic ratio (AR, high vs low, 0.5 cut, n=94). Event-free survival of patients har-192 boring a FLT3-ITD undergoing allogeneic HSCT according to (A) the mutational status of FLT3-TKD 193 and (B) according to the mutational status of FLT3-TKD and the FLT3-ITD AR (high vs low, 0.5 cut, 194 n=89).

10.   Line 207, “…prognostic information in remission: All six patients for which…” the “All” should not be capitalized.

11.   Line 245, reference(s) should be provided at the end of this statement regarding clinical studies: Subsequently, there may be a potential clinical benefit in adding the CD33 inhibitor Gemtuzumab ozogamizin also in patients with a high FLT3-ITD allelic ratio, especially when they harbor an additional NPM1 mutation, a strategy that is currently evaluated in clinical studies.

Reviewer 2 Report

This is a single center retrospective analysis investigating a series of pts with FLT3mut AML consolidated in first CR with allogeneic transplant. This reviewer has the following suggestions:

-       The abstract is somewhat misleading. The conclusion must be that MRD status before allo tx seems to be crucial – but the manuscript does not add anything that allo tx is contributing a benefit (in the absence of a comparator arm) given that this is a single arm study of a highly selected group of pts (“self-fulfilling prophecy”).

-       In line with this: A more critical discussion with a FLT3-TKI maintenance strategy based on MRD status (as an alternative to allo tx) would be intellectually desirable.

-       20% of the FLT3-mut pts were excluded from the analysis; please, explain.

-       Conditioning varied widely; this obvious bias should be discussed. This was not a homogenously treated cohort.

-       Only 5/120 pts received a TKI during induction; this is clearly not current standard-of-care. This limitation of the study (for current AML treatment) should be acknowledged and precludes definite conclusions.

-       The presence of concurrent NPM1mut does not seem to have affected the decision to allocate pts to/not to allo tx. This is not in line with current standards.

-       Figure 3: the numbers of the groups are … partially very small. Any conclusions of groups (n=2 pts) are …. risky.

Reviewer 3 Report

The presence of an FLT3-ITD and levels of the FLT3-ITD allelic ratio have been described as one of the most powerful prognostic factors in acute myeloid leukemia (AML).

Maybe just as important or even more important is present the measurable residual disease (MRD), which was already found to be an important biomarker in AML and used for prognostic, predictive, monitoring, and efficacy-response assessment.  What is more important positive MRD positivity after completion of consolidation chemotherapy, and/or MRD relapse is associated with disease relapse and inferior outcomes. Most importantly, the effect of MRD status before HSCT may also be dependent on the ELN risk category. For example, in a retrospective cohort of 176 patients with AML, Jentzsch et al. identified MRD-positive status before HSCT as a significant factor for relapse in the ELN-favorable and intermediate groups, but not in the adverse group.

But we still do not know how the FLT3-ITD allelic ratio impacts patients’ outcomes when receiving an allogeneic hematopoietic stem cell transplantation.

1.     The presented results show that before bone marrow transplantation, a greater number of FLT3-ID high AR patients had detectable residual disease.  Therefore, in my opinion, the authors should add a multivariate analysis that will answer the question of whether the mutation high FLT3-ID AR  affects the success of the transplant or whether the lack of elimination of the residual disease is the main cause fails.

2.     Moreover, including patients treated with FLT inhibitors (midostaurin or gliteritinib) tot he one group and showing survival or impact on the transplant together seems inappropriate.  In the Ratify study, the group of patients receiving midostaurin had significantly better overall survival compared to patients who did not receive it, in both groups, those who received and those who did not receive a transplant.

Round 2

Reviewer 2 Report

I believe that the authors have adequately responded.

Author Response

We thank the reviewer for his/her evaluation.